# Assessing the Role of Lipids in the Molecular Mechanism of Membrane Proteins

**DOI:** 10.3390/ijms22147267

**Published:** 2021-07-06

**Authors:** Léni Jodaitis, Thomas van Oene, Chloé Martens

**Affiliations:** Center for Structural Biology and Bioinformatics, Université Libre de Bruxelles, 1050 Brussels, Belgium; leni.jodaitis@ulb.be (L.J.); thomas.van.oene@ulb.be (T.v.O.)

**Keywords:** membrane protein, lipid–protein interaction, cryo-electron microscopy, hydrogen–deuterium exchange mass spectrometry, native mass spectrometry, single-molecule Förster resonance energy transfer, double electron–electron resonance, native mass spectrometry

## Abstract

Membrane proteins have evolved to work optimally within the complex environment of the biological membrane. Consequently, interactions with surrounding lipids are part of their molecular mechanism. Yet, the identification of lipid–protein interactions and the assessment of their molecular role is an experimental challenge. Recently, biophysical approaches have emerged that are compatible with the study of membrane proteins in an environment closer to the biological membrane. These novel approaches revealed specific mechanisms of regulation of membrane protein function. Lipids have been shown to play a role in oligomerization, conformational transitions or allosteric coupling. In this review, we summarize the recent biophysical approaches, or combination thereof, that allow to decipher the role of lipid–protein interactions in the mechanism of membrane proteins.

## 1. Introduction

Integral membrane proteins (IMPs) comprise an important part of the known human proteome [1] and have long been under scrutiny because of their role in fundamental metabolic processes, making them attractive drug targets [2]. IMPs are responsible for transmembrane transport, signaling, cell homeostasis and energy transduction, and this enumeration is not exhaustive. They are challenging to study at the experimental level because of their location, embedded in the biological membrane. Detergent solubilization is widely used to extract IMPs from the membrane but this procedure usually strips IMPs from their molecular partners: lipids. The surrounding lipids are an integral part of the machinery that allows a membrane protein to perform its function. Functional modulation of membrane proteins by lipids was established more than thirty years ago [3,4]. Membrane bulk properties, such as fluidity and lateral pressure, have been shown to affect the function of transporters, channels and other IMPs [5,6,7]. Several methods are available to measure how a specific lipid species or lipid composition affects the function of IMPs. For example, the activity of channels can be determined with electrophysiology measurements in planar lipid bilayers, and that of transporters can be assessed using reconstitution in proteoliposomes with an ionic gradient [8]. However, the molecular details of this functional modulation are a challenge to capture. The progress achieved in structural biology of membrane proteins in the last 20 years [9] yielded snapshots of lipids within high-resolution structures. These observations have triggered a renewed interest into the role of lipid–protein interactions. Other methods are now meeting the challenge to pinpoint important lipid–protein interactions and decipher their role at the molecular level. Commonly used methods to study lipid–protein interactions at the molecular level are X-ray crystallography and cryogenic electron microscopy (cryo-EM) for structural resolution, nuclear magnetic resonance (NMR), electron paramagnetic resonance (EPR), single-molecule Förster Resonance Energy Transfer (sm-FRET) and hydrogen–deuterium exchange mass spectrometry (HDX-MS), to study their dynamics, and native MS for identification [10] (Table 1). These techniques often rely on the use of a membrane mimic to study the protein of interest within a lipid bilayer. Several reconstitution systems are available for this purpose, such as proteoliposomes, bicelles, nanodiscs [11] and, more recently, saposin discs [12] and peptidiscs [13,14]. The use of styrene maleic-acid lipid particles (SMALPs)—also called ‘native nanodiscs’ or lipodisqs—is also popular since it retains the native lipid composition of the IMP under study [15]. The pros and cons of each of these mimics have been reviewed elsewhere [16].

In this review, we give an overview of how lipid–protein interactions can shape the mechanisms of IMPs at the molecular level, and the experimental approaches to study these interactions. Such modulation of the molecular mechanism can happen in various ways, but specific mechanisms have now been observed and confirmed; lipids can bind at the interface between proteins and promote or hinder their oligomerization. They can stabilize specific structural intermediates and shift the conformational equilibrium between conformers. They can directly (competition) or indirectly (allostery) modulate ligand binding (Figure 1). A combination of all these mechanisms is likely happening in any living cell. The challenge lies in disentangling these effects experimentally. This review is structured in three sections that each presents a mechanism of lipid–protein interaction, namely, oligomerization, conformational regulation and modulation of ligand binding. Each section summarizes recent studies and details the methods used to identify the role of lipid–protein interactions in the mechanism of the protein(s) under study. This review is not extensive but aims to illustrate the complexity of lipid–protein interactions with select examples highlighting the pros and cons of different experimental approaches. The studies presented here feature mostly three biophysical techniques that have changed the landscape of structural biology in the last five years: cryo-EM, structural MS and single-molecule FRET. Research works using more established structural techniques (e.g., X-ray and NMR) will be mentioned rather than detailed. Other excellent reviews on lipid–protein interactions have covered these aspects [17,18,19]. Similarly, and for the sake of conciseness, this review focuses on experimental approaches and will not detail the advances enabled by molecular dynamics (MD) simulations. Select examples of studies combining MD simulations and biophysical approaches will be illustrated in the following sections. An excellent and comprehensive review by Corradi and colleagues summarizes how MD simulations have provided countless insights into the molecular role of lipids [20].

## 2. Lipids as Regulators of Protein Oligomerization and Complex Assembly

The role of lipids in the oligomerization of membrane protein complexes is well known [22,23] but the methods to observe such a mechanism vary widely. High-resolution structures of oligomeric IMPs occasionally capture lipids at the interface of the monomeric unit of oligomers [24,25,26,27,28,29,30,31,32,33]. In 1998, a crystal structure of bacteriorhodopsin revealed a lipid molecule between each monomer as well as a lipid patch in the central cavity [28]. In 2002, the structure of KcsA, a homotetrameric channel, was solved by X-ray crystallography and displayed the presence of a lipid between each monomer [34] (Figure 1). In 2013, BetP, an osmotic stress-regulated betaine transporter, was crystallized in complex with anionic lipids [27]. Central lipids bound within the trimer suggest that lipids are used as cofactors during trimeric assembly (Figure 1). These three IMPs belong to vastly different families and have different architectures, pointing towards the ubiquity of such phenomena. Most structures of the membrane proteins solved by X-ray crystallography are carried out in the presence of a detergent, which often disrupts oligomeric assemblies. Detergent molecules also compete for lipid-binding sites, hindering native lipid binding and potentially disrupting oligomers.

These limitations are now overcome with the advent of cryo-EM, that allows structure resolution in membrane mimics. For example, a recent study by Sun and colleagues illustrates the role of lipids in maintaining the multimeric assembly of ACIII [35], an actor of the photosynthetic electron transport chains of many bacteria. The whole complex was purified using SMALPs to retain the native lipids environment. The authors obtained a structure of the complex with eleven ordered phospholipid molecules. Most of the eleven lipids are found at protein–protein interfaces, between ActC and ActF subunits. The structure also revealed a triacylated cysteine residue at the N terminus of ActB, with two other phospholipids located just next to this lipid anchor. The authors hypothesize that the lipids are necessary to keep the complex together and in the membrane, a suggestion supported by MD simulations. Another multi-unit complex, the F_0_/F_1_ ATP synthase from *E. coli*, also presents lipids located at specific regions. Sobti and colleagues identify lipid densities bridging the rotor stator interface and speculate that the lipids have a role in maintaining an energetically optimal subunit arrangement [36]. The TRPV1 structure resolved in nanodiscs in the presence of a bivalent tarantula toxin reveals a tripartite complex wherein membrane lipids form bridging interactions between the toxin and channel. Hydrophobic fingers of the toxin insert through the outer leaflet of the bilayer [37]. Thus, lipids can really act as a glue that keeps a complex IMP together. As more and more high-resolution structures in membrane mimics are solved, it can reasonably be expected to observe more of such interactions.

Another technique that has helped revealing the role of lipids in oligomerization of IMPs is native mass spectrometry [38,39,40,41] (nMS). nMS retains non-covalent interactions of biological complexes pre-existing in solution in the gas phase. It is ideally suited to observe protein–lipid complexes, without the need for a high-resolution structure. Using this technique, Gupta and colleagues demonstrated elegantly the direct role of lipids in regulating oligomerization [38]. The authors ranked all known alpha-helical IMPs structures based on their computationally predicted oligomeric stability. They then performed nMS experiments on dimeric proteins predicted to have a weak oligomeric interface, LeuT and NhaA. The authors observed that the measured mass of the dimeric form was systematically higher than the summed mass of the two monomers. Blasting the complex apart in a mass spectrometer revealed that the mass excess was caused by cardiolipin (CL) sticking to the dimer. The authors then combined mutagenesis on the oligomeric interface, delipidation and relipidation experiments and the use of a CL-deficient expression strain to demonstrate that cardiolipin is required for dimerization of LeuT and NhaA. Another study that compared the role of lipids on the mechanism of sodium antiporters from different hosts confirmed the requirement of CL for NhaA dimerization [42]. A systematic nMS study by Reading et al. explored the effects of different lipid species on the oligomerization properties of the mechanosensitive channel MscL [43]. The equilibrium between pentameric and tetrameric oligomers was mostly dependent on the detergent used for nMS studies, but lipids fine-tuned such dependence. It is interesting to note that some lipids could act either as a stabilizer of the native pentameric form or destabilizer favoring lower oligomeric forms, depending on the temperature used for lipid incubation and the detergent used for transfer into the gas phase. This dual effect illustrates the exquisite sensitivity to the variables regulating oligomerization. In another nMS study, Pyle et al. established the requirement for specific lipids for the formation of a functional dimer of the eukaryotic purine transporter UapA [41]. nMS measurements of a delipidated UapA showed that the transporter was mainly present in its monomeric form. The restoration of the dimeric form was obtained by addition of the phospholipids phosphatidylethanolamine (PE) and phosphatidylinositol (PI) (Figure 2). The authors observed an additive effect of the two different phospholipids. In contrast, addition of phosphatidylcholine disrupts the dimeric interface, demonstrating the specificity of the mechanism. Another example of lipid-mediated dimerization is that of the transporter NHE9. The authors used a combination of cryo-EM structural resolution, nMS measurements, functional assays and MD simulations. They hypothesize that phosphoinositides bridge the monomer not at the dimer interface but just above, through interaction with a loop conserved through the same protein family [44].

A recent study discovered the role of negatively charged phospholipid phosphatidylinositol-4,5-biphosphate (PIP_2_) into oligomerization of the serotonine transporter SERT without relying on nMS nor structural resolution [45]. Tissue-dependent oligomerization of SERT had been observed previously [46] and another study had identified a functional modulation of SERT by PIP_2_ [47]. The authors used a specific technique of single-molecule fluorescence developed in their laboratory to look at the oligomerization kinetics of SERT in vivo, in the plasma membrane and the endoplasmic reticulum (ER). The plasma membrane contains about 1% of PIP_2_ but the lipid species is virtually absent from the ER membrane. The authors disrupt putative PIP_2_ binding sites by mutagenesis of SERT. In parallel, they co-express the receptor with a PIP_2_-specific lipase. By combining these two approaches, they can show that the serotonin receptor oligomeric state is correlated with PIP_2_ availability. These research works and others [48,49] demonstrated that lipids can be part of the subunit interface, and directly stabilize the complex assembly. These findings also explain a number of functional studies that had observed a lipid-dependent modulation of function [7,50].

While factors governing lipid-mediated oligomerization of IMPs are not yet understood, the research works presented here showcase both the versatility and importance of such interactions. Specifically, the use of nMS to determine lipid-induced oligomerization opens the door to studies of IMPs with an unknown structure. So far, nMS studies looking at the effect of lipids in oligomerization were carried out in mixed detergent–lipid micelles. Recently, the availability of MS instruments designed to bring bigger and more complex systems into the gas phase—coupled with dedicated software development for the analysis of IMPs embedded in nanodiscs (Unidec)—has facilitated nMS studies of proteins in nanodiscs [51,52] or SMALPs [53]. Such progress will lead to more research on lipid-mediated oligomerization in a detergent-free environment [54,55].

## 3. Lipids Stabilizing Conformational States

Membrane proteins are dynamic entities and have to adapt structurally during a functional cycle. For example, transporters will open to opposite sides of the membrane, whereas channels will open and close to provide a pathway for ions and GPCRs will transmit a transmembrane signal by switching between active and inactive states. These structural rearrangements happening within the membrane often imply an active role for the lipids, interacting directly or not with specific structural motifs to modulate the energy landscape. Different approaches have been used to experimentally determine the role of lipids in regulating conformational transitions. The most direct methods rely on distance measurements in membrane mimics or observation of high-resolution structures in different conformations. More indirect approaches measure the changes in solvent accessibility of the secondary structure elements or specific amino acids. Distance measurements depend on the introduction of probes, and such chemical modification of the protein under study may affect its function. In that regard, techniques aiming at looking at dynamics using accessibility are less invasive, and can be applied more broadly. Among these, solid-state NMR is now an established spectroscopic method to investigate the dynamics of membrane proteins in bilayers [57,58,59]. Its principles and applications are described in excellent reviews [60,61,62] and will not be discussed in the present work.

### 3.1. Measuring Distances

In the last five years, two techniques were mainly used to measure the nanometric distance changes that accompany conformational changes of IMPs: single-molecule Förster Resonance Energy Transfer (sm-FRET) and double electron–electron resonance (DEER also called PELDOR). Both rely on the introduction of a pair of probes on the protein, usually tethered chemically via a cysteine residue. Sm-FRET requires the introduction of a donor and an acceptor probe. The efficiency of energy transfer (E_FRET_) to the acceptor after exciting the donor probe is distance-dependent. E_FRET_ is measured by detecting the fluorescence signal of both the donor and acceptor probes, using dedicated microscopes [63]. DEER uses a pair of identic spin labels, typically the thiol-specific probe MTSSL. DEER measurements extract the spin coupling between both labels, which is also distance-dependent [64]. Both techniques have pros and cons in terms of sensitivity, equipment needed, experimental setup and analysis of the signal, detailed in excellent reviews [65,66].

Among the diverse classes of IMPs, transporters are the target of choice for smFRET and DEER studies. The motions necessary to switch from inward open to outward open are large enough to be detectable (typically 15 to 60 angstroms). Furthermore, transporters often present structural conservation from prokaryotic to human homologs. Bacterial homologs can be used as proxies for structural studies of human counterparts, often harder to produce for biophysical studies. They are also often monomeric or dimeric, which facilitates the labeling strategy, compared to multimeric channels, for example. For these practical reasons, DEER [67,68,69,70] and smFRET [71,72,73,74] studies in membrane mimics are heavily biased towards transporters.

Liu and coworkers compared the conformational transitions of the lipid flippase MsbA in detergent micelles, nanodiscs and liposomes [74]. The authors looked at the dynamics of the nucleotide-binding domains (NBDs) and the transmembrane domains (TMDs) with smFRET, using a pair of probes on each side of the transporter. They could show that the dynamics of the TMDs are restricted in a lipid environment, while the dynamics of the NBDs are larger in proteoliposomes and detergent micelles compared to nanodiscs. The transition between different states is faster in liposomes compared to the other environments. A similar comparison between detergent micelles and the lipid environment of proteoliposomes was carried out for the peptide transporter McjD [71], using smFRET. The authors show that the trigger of the conformational transition is conserved regardless of the environment; both in detergent micelles and proteoliposomes, the addition of the peptide mccj25 opens the extracellular side, while the addition of ATP closes the NBDs. Similarly, the vitamin transporter BtuCD was studied by smFRET in nanodiscs and detergent micelles. The conformational dynamics of BtuCD confirmed the general trend: while the overall features of the transport cycle are conserved in detergent micelles and nanodiscs, the extent of the conformational changes differs [72]. A similar conclusion was reached in an extensive DEER study on the secondary multidrug transporter LmrP [68]. The authors performed distance measurements on the transporter in detergent micelles and nanodiscs and identified a specific structural motif that works as a protonation-dependent conformational switch [68]. They could show that the conformational states visited in the presence or absence of lipids are similar. However, the lipid environment modulates the pK_a_ of the conformational switch and facilitates a conformational transition at a physiological pH. To go further into their molecular characterization, they use synthetic phospholipids. They demonstrate that methylation of the phosphatidylethanolamine headgroup tips the conformational equilibrium towards the outward-open state. Such dependence on the chemistry of the membrane was also observed for the transporter DtpA [73]. The authors used the Salipro system to reconstitute the peptide transporter DtpA in soluble discoidal bilayers composed either of phosphatidylethanolamine POPE, phosphatidylserine POPS, phosphatidic acid POPA or brain lipids extract. They demonstrate that anionic lipids promote a ‘very inward open state’ not observed in crystal structures of DtpA homologs. It is worth noting that for both LmrP and DtpA, the synthetic lipids used for reconstitution have the same aliphatic tail. This suggests that the headgroup is the main driver of the change in the conformational equilibrium.

The importance of the lipid environment is well illustrated in a study by Jagessar et al. on the PfMATE multidrug exporter [75]. The isomerization between the outward-facing and inward-facing states is strictly dependent on the presence of a lipid bilayer. DEER distances measurements in detergent micelles show no isomerization upon ligand binding, but similar measurements in nanodiscs captured large-scale conformational changes upon protonation of PfMATE, indicative of the conformational transition (Figure 3). Another striking example of the importance of the lipid bilayer in stabilizing specific conformations comes from the DEER study of the multidrug transporter MdfA in nanodiscs. The lipid environment allowed to isolate a doubly occluded intermediate, where the periplasmic and cytoplasmic sides are both closed, compared to published crystals structures [70]. This observation led to the hypothesis that for this transporter, the state compatible with substrate binding is closed. Lipophilic substrates come from the membrane, in line with a model suggested for homologous transporter LmrP [76]. Another DEER study of the eukaryotic ABC transporter P-glycoprotein (P-gP) in nanodiscs shows that the high-energy post hydrolysis state of the transporter is stabilized by the lipid bilayer [69]. These DEER and smFRET studies on various transporters demonstrate that molecular level interaction between the membrane and the IMP can shape the conformational landscape.

### 3.2. Looking at High-Resolution Structures

A direct way to look at how lipids modulate IMP conformations has become accessible thanks to cryo-EM. With a bit of luck, an IMP solved by cryo-EM in a lipid environment will reveal which lipids are bound where, and provide a direct read-out of their role in stabilizing a specific conformer. The technique also allows to solve different conformations present in a sample if the conformations are different enough to be sorted into different classes during the image processing workflow [77]. This has proven particularly fruitful for channels, whose important size (often >100 kDa) makes them easily amenable to cryo-EM structural resolution. In the last few years, an important number of channels has been solved in nanodiscs, such as thermo-sensitive channels (TRPM4 [26], TRPM1 [78], TRPV2 [79], TRPV3 [80,81], TRPV5 [82,83], TRPV6 [84] and TRPV1 [25]), mechanosensitive channels (MscS [85,86,87] and OSCA1.2 [88]), voltage-gated channels (TASK2 [89] and TPC1 [90]) and other channels, such as PKD2 [91] and TMEM16 [92]. Structures of a few reconstituted transporters have also been solved, such as the bacterial multidrug transporters AcrB [93] and TmrAB [94], and human multidrug transporter ABCG2 [95]. The majority of these structures have shown lipids bound to specific locations, leading to various hypotheses about their role. In the next paragraphs, we provide a few examples where the structural resolution has allowed to deduce a lipid-mediated mechanism of IMP function.

A telling example is that of the TRPV3 channel. Two research groups solved the structure of the channel in nanodiscs and their findings were published simultaneously [80,81]. Shimada and colleagues solved the structure of mouse TRPV3, while Deng and colleagues solved that of human TRPV3. Both teams captured the ligand-free, closed state. In the case of human TRPV3, a sensitization-prone mutant was used to obtain additional structures in the open and inactive states. In both studies, the structures deviate substantially from the previous ones obtained in the absence of a lipid environment [96,97,98]. The most noticeable change is that the pore in its closed state presents two constriction sites with a narrow selectivity filter, which is completely absent in the previous structures of the closed channel. In the mouse TRPV3, a phospholipid is observed in the pore. This lipid would be sterically clashing with the selectivity filter in the conducting pore, suggesting that the lipid stabilizes the narrow filter of the closed confirmation [81]. This lipid is not observed in the human TRPV3 structure, probably because the lipids used for nanodisc reconstitution are different. Both groups suggest that lipids are directly involved in the heat-sensing mechanism that activates the channel, as a sensor and relay of temperature changes.

Mechanosensitive channels were among the obvious targets for structure resolution in a lipid environment since their mechanism is directly related to lipid–protein interactions. These channels open in response to membrane tension, but whether the structural changes are caused by hydrophobic mismatch, changes in membrane curvature or anisotropic forces in the bilayer is debated [99]. Three groups solved the structure of the prototypical mechanosensitive channel MscS in nanodiscs by cryo-EM: Rasmussen et al. in 2019 [86], Reddy a few months later [85] and Zhang et al. in early 2021 [87]. The membrane-embedded MscS obtained by all three groups appears to be in the closed conformation, which is expected in the absence of tension. To test whether membrane thinning would cause the closed to open transition, Zhang and colleagues solved the structure of MscS in nanodiscs made of short-chains lipids (10-PC). The obtained structure is only partially open, demonstrating that hydrophobic mismatch is not sufficient to open the channel. The authors then incubated the reconstituted protein with cyclodextrin, to selectively remove lipids from the nanodiscs. They reasoned that the remaining lipids would have to stretch to cover the hydrophobic regions, therefore introducing tension in the membrane. The structure obtained in this bilayer under tension is that of a desensitized channel, where the transmembrane domain is collapsed, and the conducting pore is narrow. The authors then hypothesized that the completely open conformation is dynamic and transient, and hence difficult to capture. Zhang and colleagues favor it, using a mutant that does not desensitize. All the structures obtained present lipids in several key locations, which allow to deduce their role in the structural transitions. Lipids are observed inside the conducting pore in the closed state. These pore lipids act like a plug and prevent ion conduction. They move to the periphery of the pore and then out of it as the channel transitions to open and sub-conducting states, and as its transmembrane domain tilts and collapses. Additional lipids are located in grooves between the subunits and maintain the closed conformation by physically preventing sliding and tilting of the subunits to transition towards the conductive states. As the channel transitions to open and desensitized states, the pore lipids move to the periphery and then leave the pore, while the lipids leave the grooves, allowing the gradual collapse of the transmembrane domain. Thus, the motions of the lipids directly drive the structural changes and therefore function of the channel.

Another interesting effect of the lipid environment is observed for the channel LRRC8A. The structure of the channel was resolved in nanodiscs by Kern and colleagues [29]. The protein is a homohexameric channel that presents a six-fold symmetry. By comparing their structure with others obtained in the detergent digitonin, the authors observed that the lipid environment causes a change in the symmetry of the channel. LRRC8A in digitonin displays a three-fold symmetry and is formed of three pairs of asymmetric dimers. This difference in symmetry arises from the presence of lipids in a specific gap between the subunits. When the lipid environment is removed, the channel rearranges in a way that forms three smaller gaps and three larger gaps between the subunits, the large ones being filled with digitonin. Thus, the presence of lipids is required to lock the monomeric unit in a specific conformation that leads to a symmetric hexameric channel. These examples show unequivocally that IMPs work as lipoprotein complexes and their mechanism of action is dictated by lipids and lipids motions.

### 3.3. Measuring Accessibility Changes with H/D Exchange

A method that has shown a huge increase in its use to study membrane protein dynamics in the last five years is hydrogen–deuterium exchange coupled to mass spectrometry (HDX-MS) [100,101,102,103]. This method measures the rate of exchange of labile protons from the backbone amide upon incubation in a deuterated solvent, leading to mass shifts of the peptic peptides detectable by LC-MS [104]. The extent of H/D exchange is directly related to the stability of the H-bond and to its accessibility to the solvent. Deducing conformational intermediates from accessibility changes requires a structural framework, usually obtained through high-resolution structures of the IMP (or a homolog) in different conformations. Once this framework is established, changes in solvent accessibility can be correlated to specific conformational intermediates [105]. HDX-MS applied to IMPs has mostly been used to study the dynamics of GPCRs and transporters [106]. Many HDX-MS studies of GPCRs were carried out in bicelles [107,108], but these studies did not investigate the role of the lipid environment per se.

The use of HDX-MS to specifically interrogate the role of the lipid environment on the conformational dynamics was used to study the secondary transporters LeuT [109,110] as well as LacY and XylE [111], and the ABC transporters BmrA [112,113] and P-glycoprotein (P-gP). Most of these studies were carried out using the nanodisc system as a membrane mimic. The methodology was developed in 2010 by Hebling and colleagues on the model protein γ-Glutamyl carboxylase (GGCX) [114]. In 2017, this method was adapted by Adhikary et al. to study the neurotransmitter homolog LeuT [115,116]. The authors reconstituted the protein in the nanodiscs in either outward-favoring or inward-favoring conditions and mapped its overall conformational hallmarks. Aided by MD simulations in a bilayer, they pinpoint which secondary structure elements display a specific dynamic behavior depending on the conformation. Importantly, this research work illustrates how HDX-MS can offer a global view of IMP dynamics in a bilayer. Atkins and colleagues reconstituted the human multidrug transporter P-gP in nanodiscs and compared its HDX-MS profile with that of the protein in detergent micelles [117]. They found an overall similarly rough conformational landscape in both cases, but information is mostly limited to the NBDs because the sequence coverage of the TMDs is low. They nevertheless observed asymmetry in the exchange behavior between the NBDs, thereby confirming, with a non-invasive methodology, this observation, which was also made with DEER [69]. Another study on the secondary transporters from the Major Facilitator Superfamily (MFS) delves deeper into the molecular details of lipid–protein interactions. Martens and colleagues compared the conformational states of the secondary transporters XylE and LacY in nanodiscs of different lipid compositions comprising or not the main bacterial phospholipid PE [111]. They use MD simulations in bilayers matching the composition of the nanodiscs to identify a specific PE–protein interaction. They carry out HDX-MS experiments in nanodiscs on the WT and on proteins mutated at the site of the predicted interaction. They show that PE favors the inward open conformation trough electrostatic interaction with a conserved conformational switch (Figure 4). Another notable methodological improvement was the use of SMALPs for HDX-MS experiments, allowing to study IMPs in a native-like environment. Reading and colleagues prepared SMALPs of the rhomboid protease GlpG in three different native lipid environments, obtained through the use of either different bacterial strains or different growth conditions [118]. The difference in the lipid compositions (evaluated by lipidomics) could be linked with differences in local dynamics proving the versatility of the method.

These examples show that HDX-MS is a useful tool to follow changes in conformational dynamics of reconstituted IMPs. Since its use is facilitated by the availability of high-resolution structures, it is poised to become a standard tool for the structural biology of IMPs in native-like environments.

## 4. Lipids as Modulators of Ligand Binding

Lipid molecules can occupy specific sites within IMPs, either competing directly with ligand binding by modulating the drug-binding pocket [119] or causing long-range allosteric effects [120]. Detecting such effects typically requires a combination of high-resolution structural data and functional data on the mutants compromising lipid-binding sites [121,122,123]. A textbook example of allosteric coupling by a lipid molecule was provided by the structural resolution of the TRPV1 channel in nanodiscs [37]. The authors observed a shared binding pocket for phosphatidylinositol (PIP) lipid and the agonist resiniferatoxin (RTX). When the agonist is not present, the binding pocket is occupied by PIP. Binding of RTX kicks PIP from the pocket and by doing so changes the coordinating residues. This small structural rearrangement amplifies by pulling the S4–S5 linker away from the central axis thereby facilitating opening of the lower gate to activate the channel. Another recent example of allosteric modulation by a lipid molecule comes from a study of ELIC, a pentameric ligand-gated ion channel (pLGIC)**.** Functional modulation of pLGICs by lipids is well documented [124,125]. The structure of the anionic channel was resolved by cryo-EM in nanodiscs and reveals a phospholipid molecule located near a conserved proline kink in transmembrane helix M4. The authors identify this kink as a conserved feature in anion selective channels and GABA receptors. The mobility of the helix couples ligand binding to channel opening and desensitization. A combination of mutagenesis and electrophysiology measurements show that occupation of this site either by a lipid molecule or a drug stabilizes a closed conformation [126]. Competitions between an exogenous drug an endogenous lipid appears to be an important mode of regulating channel opening, and is suggested for other IMPs (GABA receptor [127], serotonin receptor [119] and voltage-gated channel TPC1 [90]).

Another good example of the impact of lipid–protein interactions on allosteric modulation is provided by an interesting study on the A2AR receptor, which does not involve structure resolution. Radioligand-binding assays performed by Guixà-González et al. show that binding of cholesterol significantly reduces the binding of the antagonist to the receptor [128]. Subsequent MD analysis predicted direct entry of cholesterol from the membrane into the orthosteric binding site. The authors designed an elegant assay to confirm the MD predictions. The presence of a cholesterol molecule inside the receptor, clashing with the orthosteric site, would prevent labeling of cysteine residues with a reactive probe. Sequential addition of the probe and removal of cholesterol in the presence and absence of the antagonist strongly suggests that cholesterol is inside the receptor. The authors demonstrate that the observed inhibitory effect of cholesterol was not only due to allosteric changes (as previously shown for this GPCR and others [129]), but also to direct occupation of the orthosteric binding site. This research opens the way to the potential use of sterol and sterol-like compounds in GPCR therapeutics.

Another tool that facilitates the identification of coupled interactions between lipids and ligands or cofactors is native MS [130]. The order of incubation of the ligands/cofactors allow to determine whether drug binding correlates positively or negatively with lipid binding. One of the first studies showing such synergy was carried out by Marcoux et al., on the multidrug efflux pump P-glycoprotein [131]. Preincubation of the transporter with cardiolipin (CL) before addition of ligand cyclosporin A (CsA) had no effect on CsA binding. However, when CsA was added prior to CL incubation, an increase in lipid binding was observed. This suggests that CsA binding primes the transporter for subsequent lipid binding, either through a conformational change or adaptation of the ligand pocket. Such a method was used successfully on GPCRs [132], channels [133,134,135,136], transporters [131,137,138,139] and other IMPs [140], highlighting the role of specific lipid species in allosteric coupling.

A recent article combines both high-resolution structure determination, nMS and functional tests to demonstrate that the presence of a lipid can modulate polyspecificity of the bacterial multidrug transporter LmrP (Figure 5). The authors obtained the crystal structure of the protein in a ligand-bound state. An unknown density observed close to the ligand shows the features of a phospholipid molecule. Molecular dynamic simulations identify the phosphatidylglycerol (PG) molecule as the more likely candidate to stabilize the ligand inside the binding pocket. To confirm the presence of a PG molecule inside the binding pocket of LmrP, the authors performed nMS experiments in PG-containing nanodiscs. They compared the spectrum of WT LmrP and a mutant, designed to compromise lipid binding, at different voltages. They showed that a PG molecule remained attached to the WT protein at high voltages, but the mutant did not present such preferential binding. These measurements demonstrated a specific affinity for PG. Additional transport assays confirmed the importance of keeping the lipid-binding site intact for the efflux of some but not all ligands. The authors hypothesize that the lipid provides a malleable hydrophobic environment for diverse substrates, providing a rationale for the substrate polyspecificity of the multidrug transporter.

## 5. Conclusions and Perspectives

Membrane proteins work within a complex lipidic environment that participates in the molecular mechanism enabling function. It is becoming clear that membrane proteins have to be considered as lipoprotein complexes. The methods and approaches to study them are adapting to that complexity (Table 1). Recent work has shown that different lipid species can have different albeit interlinked roles in the mechanisms of IMPs [141,142]. For example, studies on the Na, K-ATPase have highlighted distinct roles for distinct lipid species. Two X-ray structures have identified two lipid-binding sites at different locations [143,144] and a third study has determined the role of lipids at these sites using native MS, transport assays and mutagenesis. Cholesterol and PS stabilize the protein without affecting its activity while PC or PE stimulate activity by accelerating the conformational transitions, but do not affect stability [145]. Integrative approaches that combine biochemical and biophysical characterization with high-resolution structures are becoming standard and provide updated structure–function relationships of the IMPs.

As our understanding of membrane proteins shifts towards describing lipoprotein complexes, other layers of complexity are now added into the equation. Eventually, obtaining molecular information in the cellular context is the next aim of structural biology. Techniques that report molecular interactions in vivo or in cellulo are fast developing. Cryo-electron tomography [146] and super resolution microscopy techniques [147] are making strides towards molecular-level resolution. Native mass spectrometry of whole cells has yielded novel insights into the organization of IMPs in their truly native membrane [148]. Not only the structural but also the temporal resolution of biophysical techniques are increasing [149,150,151], and the possibility to look at IMPs in action is getting closer. To this end, IMPs activated by ion gradients need the right environment. Yao and colleagues have devised a protocol to carry out cryo-EM experiments in proteoliposomes, using the prototypical AcrB MDR efflux pump as a model. Such a development paves the way for further studies of functioning IMPs [152]. The future looks bright and exciting for the field of structural biology of membrane proteins, with many methodological improvements making the mysteries of the cell visible to the human eye.

## Figures and Tables

**Figure 1 ijms-22-07267-f001:**
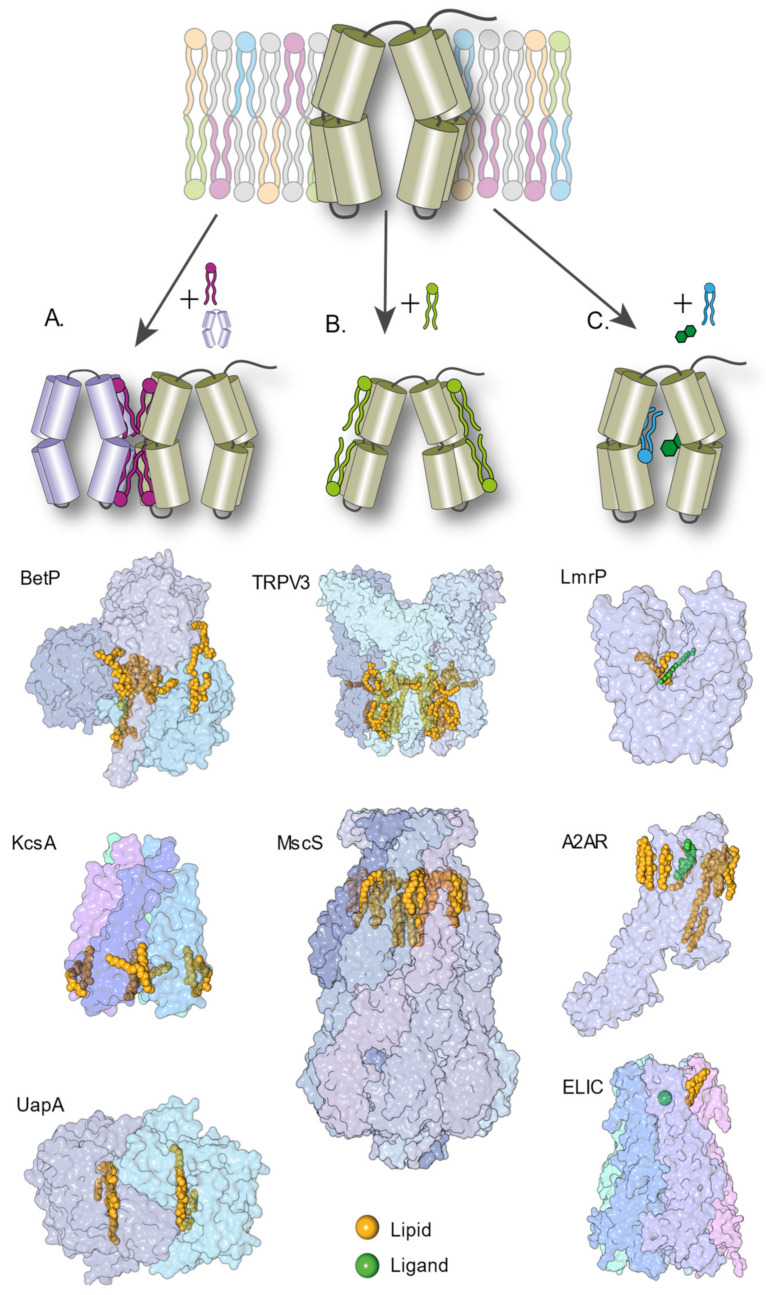
Schematic representation of known molecular mechanisms mediated by lipids and representative examples of lipid–protein complexes captured in high-resolution structures (**A**). Lipids as regulators of protein oligomerization and complex assembly. Cartoon representation of BetP (PDB ID 4C7R), KcsA (PDB ID 3IFX) and UapA (PDB ID 5I6C) proteins. (**B**). Lipid regulating conformational dynamics. Cartoon representation of TRPV3 (PDB ID 6LGP) and MscS (PDB ID 6PWN) proteins (**C**). Lipids modulating ligand binding directly (competition) or indirectly (allostery). Cartoon representation of LmrP (PDB ID 6T1Z), A2AR (PDB ID 5IUA) and ELIC (PDB ID 6HJX) proteins. PDB figures were generated using ‘The Protein Imager’ [21].

**Figure 2 ijms-22-07267-f002:**
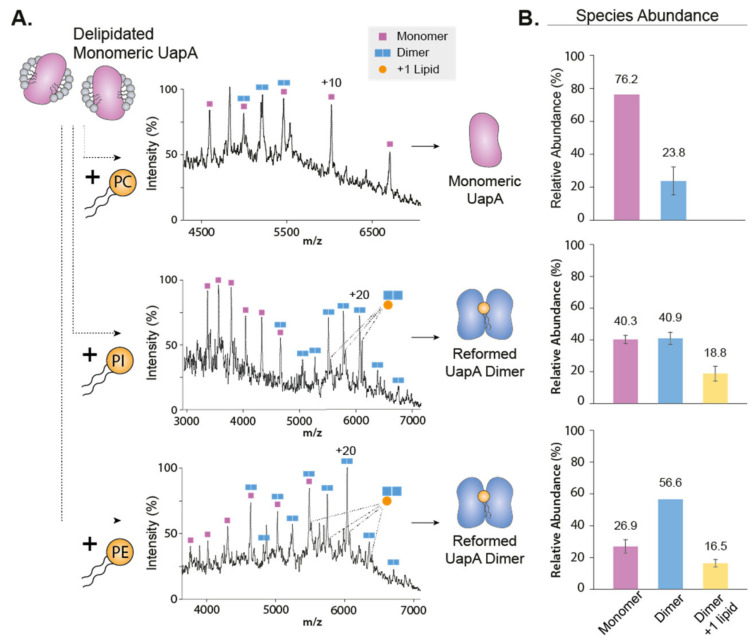
Addition of lipids to delipidated UapAG411V_Δ1-11_ reforms the dimer. (**A**) Mass spectra showing the effects of PC (34:1, upper), PI (34:1, middle) and PE (34:1, lower) on the oligomerization of delipidated UapAG411V_Δ1-11_. Peaks of the lipid-bound species are highlighted. (**B**) Relative abundances of monomer and dimer species in the presence of PC (upper), PI (middle) and PE (lower) were quantified using UniDec software [56]. Adapted with permission [41].

**Figure 3 ijms-22-07267-f003:**
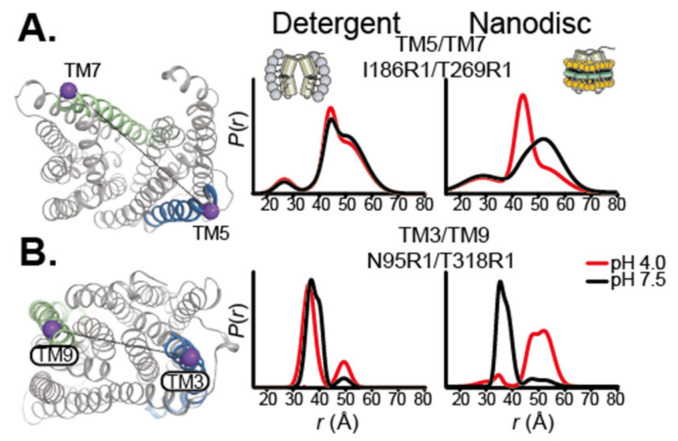
Ligand-dependent conformational dynamics of PfMATE require a lipid environment. (**A**) Representative spin label pair sampling distances between TM5 and TM7 on the extracellular side and (**B**) TM3 and TM9 on the intracellular side of PfMATE. The spin label locations are highlighted on the OF structure by purple spheres connected by a line. The helices targeted in the N-lobe and C-lobe are highlighted in blue and green, respectively. Distance distributions, representing the probability of a distance P(r) versus the distance (r) between spin labels, are shown in black traces at pH 7.5 and red traces at pH 4.0 in DDM micelles (**left**) and lipid nanodiscs (**right**). Adapted with permission from [75].

**Figure 4 ijms-22-07267-f004:**
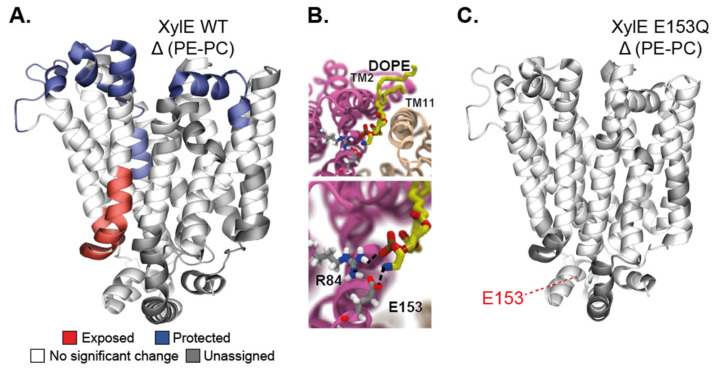
Lipid–protein interactions regulate the conformational equilibrium. (**A**) Differential deuterium uptake pattern (ΔHDX) of WT XylE in DOPE-PG-CL nanodiscs (native-like) minus DOPC-PG-CL (control) mapped onto the 3D structure of XylE (PDB: 4GBY). Red- and blue-colored regions indicate segments containing peptides with a positive ΔHDX (red—more deuteration) or negative ΔHDX (blue—less deuteration), respectively; white regions indicate that no significant ΔHDX is observed (*p* ≤ 0.01), and gray indicates regions where peptides were not obtained for both the mutant and the WT conditions. (**B**) Representative MD snapshot of the close-up of the conserved, charged residues interacting with the PE headgroup of the phospholipid. Polar interactions with R84 and E153 prevent network formation and steric hindrance prevents contacts of the TM2 and TM11. (**C**) Mutagenesis of E153 abolishes a lipid-induced conformational shift. ΔHDX of XylE in PE:PG:CL nanodiscs (native-like) minus PC:PG:CL (control) nanodiscs mapped on the PDB structure. Adapted from [111].

**Figure 5 ijms-22-07267-f005:**
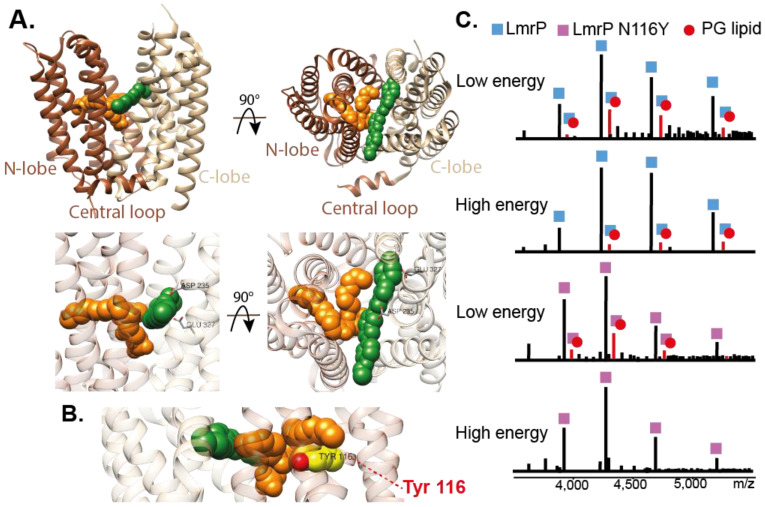
Structure of LmrP in complex with Hoechst 33342. (**A**) Cartoon representation of LmrP with the ligand Hoechst 33342 (green) and phosphatidic acid (orange) modeled from the densities observed (PDB ID 6T1Z). Hoechst 33342 forms polar interactions with D235 and E327 located in the C-lobe of LmrP. (**B**) Close up on the phospholipid. N116Y mutation perturbs lipid binding (**C**). nMS spectra showing compromised PG binding on the N116Y mutant. At a low activation energy (160 V), multiple lipids are bound to LmrP, including a peak corresponding to a single bound DOPG (in red). At high energy (200 V), this peak is still present. In the case of the N116Y mutant, although DOPG binds at low activation energy, it disappears at a high activation energy.

**Table 1 ijms-22-07267-t001:** Representative biophysical techniques used to study the lipid–protein interactions of IMPs.

Biophysical Techniques		Information Obtained
**X-ray crystallography** extrapolates the position and arrangement of atoms in single crystals from the diffraction pattern of X-ray.	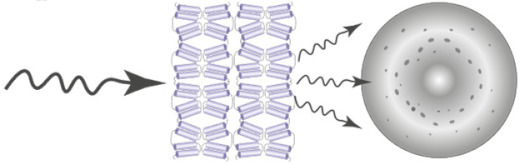	High-resolution structure.
**Cryo-electron microscopy** images flash frozen protein solutions with an electron beam. Single particles are aligned and classified in two-dimensional classes. The 3D structure is determined by reconstruction software.	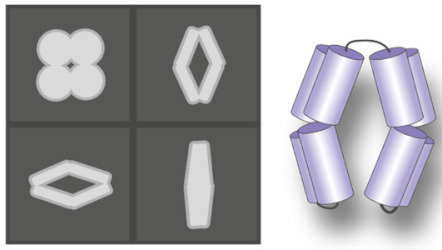	High-resolution structure.
**Native mass spectrometry (nMS)** retains in the gas-phase non-covalent interactions pre-existing in solution. Protein-protein and protein-ligand complexes can be observed.	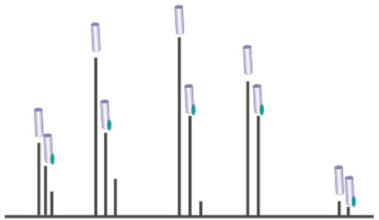	Protein–protein interactions, protein–ligand interactions.
**Hydrogen–deuterium exchange (HDX) MS** measures the rate of deuterium exchange of labile protons of the backbone amide. The exchange depends on stability of the H-bond and solvent accessibility.	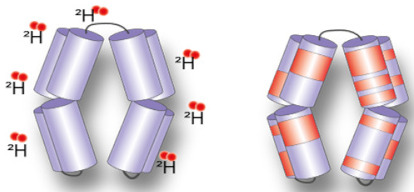	Structural dynamics, protein–protein interactions, protein–ligand interactions.
**Single-molecule FRET** measures the amount of energy transfer between a pair of fluorophores, a donor and an acceptor to perform small distances measurements (1–10 nanometers).	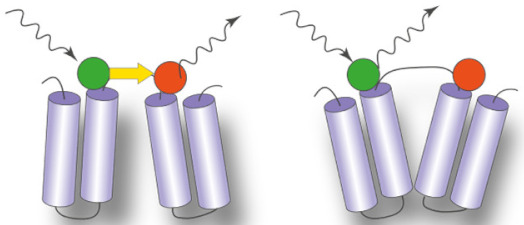	Conformational changes, kinetics.
**Double electron–electron resonance** measures the dipolar spin coupling between a pair of spin labels. The modulated spin echo contains distance information.	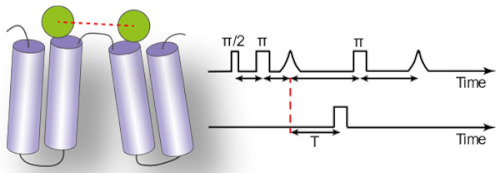	Conformational changes.
**Electron paramagnetic resonance (cw-EPR)** provides a read-out of the environment and the mobility of a paramagnetic probe within a protein, typically a nitroxide spin label.	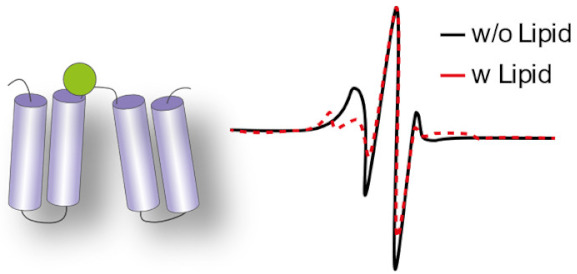	Mobility, environment, tertiary fold.
**Multidimensional nuclear magnetic resonance (NMR)** provides details about the local molecular environment of the nuclei of an isotopically labelled protein, allowing to deduce chemical bonds, distance and relative motions between nuclei.	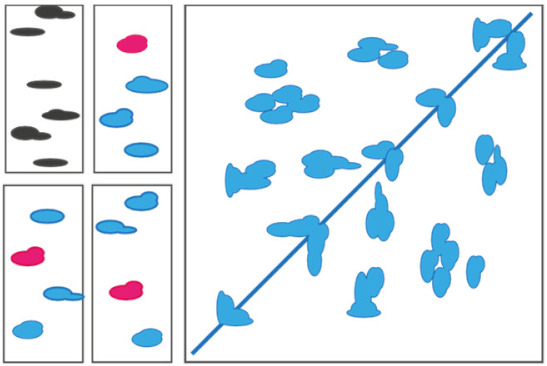	High-resolution structure, dynamics, protein–protein interactions, protein–ligand interactions.

## Data Availability

Not applicable.

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
