# Peer review of "Assessing the Role of Lipids in the Molecular Mechanism of Membrane Proteins"

_ijms, 2021, doi:10.3390/ijms22147267_

Round 1

Reviewer 1 Report

In their manuscript the authors nicely summarize recent methodological developments in the biophysical characterization of membrane-protein-lipid interactions, with a particular focus on mass spectrometry approaches, while also discussing smFRET, DEER and cryo EM. The manuscript thus presents a solid review, worthy of publication, to which I only have one minor and rather cosmetic suggestion - while I can perfectly understand the authors reasoning not to discuss the two major structural biological methods of X-ray chrystallography and NMR spectroscopy in their manuscript, I do in fact think that NMR spectroscopy (in particular solid-state NMR) and potentially also EPR should be included in Table 1 of the manuscript for comparative reasons, especially since both methods are mentioned in the manuscript directly prior to the reference to Table 1. 

Author Response

We thank the reviewer for their feedback. We have now expanded table 1 to include solid state NMR and continuous-wave EPR.

Reviewer 2 Report

In this review, the author gave an outline of the recently-emerged techniques that can evaluate specific interactions between membrane proteins and lipids with referring the relating impressive experimental results in this field. This review article is clarified in three categories of techniques that can evaluate impacts and functions of lipids on (I) control of membrane protein oligomerization, (II) conformational regulation of membrane proteins, and (III) modulation of ligand binding to membrane proteins. Its cogent classification should be helpful to understand for readers even who does not specialize in this field. Many topics, new techniques, and the related results were well-summarized clearly without bias, and therefore I recommend acceptance for your journal after minor revisions. There are several grammatical errors in the article. It should be modified.

Introduction section

  1. line 33; comma between the superscripted 3 and 4 is necessary.
  2. line 41; “these last 20 years” could be “in the last 20 years” or “in these 20 years”.
  3. line 53; no definition of “SMA”. The author should add the original technical words before abbreviation.
  4. line 73-76; there are problems in grammar of this sentence.

Section 1

  1. line 92; maybe “already” should be removed.
  2. line 161; no definition of “SERT”. The author should add the original technical words before abbreviation.

Section 2

  1. line 196; “mimics” should be “mimics” (“i” should not be in a bold-font)

Section 2.1

  1. line 215; no definition of “MTSSL”. The author should add the original compound name before abbreviation.
  2. line 245; As following the author’s rule, “LmrP” should be in a bold-font.

Section 2.3

  1. line 361; References should be put after “(HDX-MS)”.
  2. line 373-374; As following the author’s rule, “LeuT”, “LacY”, “XylE”, “BmrA”, and “P-gP” should be in a bold-font.
  3. line 377; maybe, “and adapted” should be removed.
  4. line 378; References should be put after “LeuT”.

Section 3

1. line 463; “(Figure 5)” should not be in a bold-font. 

Author Response

We thank the reviewer for their careful reading of the manuscript. We have implemented all the suggested changes and added missing references.